# Candidate Modifier Genes for the Penetrance of Leber’s Hereditary Optic Neuropathy

**DOI:** 10.3390/ijms231911891

**Published:** 2022-10-06

**Authors:** Hui-Chen Cheng, Sheng-Chu Chi, Chiao-Ying Liang, Jenn-Yah Yu, An-Guor Wang

**Affiliations:** 1Program in Molecular Medicine, College of Life Sciences, National Yang Ming Chiao Tung University, Taipei 11221, Taiwan; 2Department of Ophthalmology, Taipei Veterans General Hospital, 201 Sec. 2, Shih-Pai Rd., Taipei 11217, Taiwan; 3Department of Ophthalmology, School of Medicine, National Yang Ming Chiao Tung University, Taipei 11221, Taiwan; 4Department of Life Sciences and Institute of Genome Sciences, College of Life Sciences, National Yang Ming Chiao Tung University, Taipei 11221, Taiwan; 5Brain Research Center, National Yang Ming Chiao Tung University, Taipei 11221, Taiwan; 6Department of Ophthalmology, Taichung Veterans General Hospital, Taichung 40705, Taiwan

**Keywords:** Leber’s hereditary optic neuropathy, LHON, whole exome sequencing, nuclear modifier genes

## Abstract

Leber’s hereditary optic neuropathy (LHON) is a maternally transmitted disease caused by mitochondria DNA (mtDNA) mutation. It is characterized by acute and subacute visual loss predominantly affecting young men. The mtDNA mutation is transmitted to all maternal lineages. However, only approximately 50% of men and 10% of women harboring a pathogenic mtDNA mutation develop optic neuropathy, reflecting both the incomplete penetrance and its unexplained male prevalence, where over 80% of patients are male. Nuclear modifier genes have been presumed to affect the penetrance of LHON. With conventional genetic methods, prior studies have failed to solve the underlying pathogenesis. Whole exome sequencing (WES) is a new molecular technique for sequencing the protein-coding region of all genes in a whole genome. We performed WES from five families with 17 members. These samples were divided into the proband group (probands with acute onset of LHON, *n* = 7) and control group (carriers including mother and relative carriers with mtDNSA 11778 mutation, without clinical manifestation of LHON, *n* = 10). Through whole exome analysis, we found that many mitochondria related (MT-related) nuclear genes have high percentage of variants in either the proband group or control group. The MT genes with a difference over 0.3 of mutation percentage between the proband and control groups include AK4, NSUN4, RDH13, COQ3, and FAHD1. In addition, the pathway analysis revealed that these genes were associated with cofactor metabolism pathways. Family-based analysis showed that several candidate MT genes including METAP1D (c.41G > T), ACACB (c.1029del), ME3 (c.972G > C), NIPSNAP3B (c.280G > C, c.476C > G), and NSUN4 (c.4A > G) were involved in the penetrance of LHON. A GWAS (genome wide association study) was performed, which found that ADGRG5 (Chr16:575620A:G), POLE4 (Chr2:7495872T:G), ERMAP (Chr1:4283044A:G), PIGR (Chr1:2069357C:T;2069358G:A), CDC42BPB (Chr14:102949A:G), PROK1 (Chr1:1104562A:G), BCAN (Chr 1:1566582C:T), and NES (Chr1:1566698A:G,1566705T:C, 1566707T:C) may be involved. The incomplete penetrance and male prevalence are still the major unexplained issues in LHON. Through whole exome analysis, we found several MT genes with a high percentage of variants were involved in a family-based analysis. Pathway analysis suggested a difference in the mutation burden of MT genes underlining the biosynthesis and metabolism pathways. In addition, the GWAS analysis also revealed several candidate nuclear modifier genes. The new technology of WES contributes to provide a highly efficient candidate gene screening function in molecular genetics.

## 1. Introduction

Leber’s hereditary optic neuropathy (LHON) is a maternally transmitted disease characterized by acute and subacute visual loss predominantly affecting young men [1,2,3]. It usually onsets between 15 and 35 years of age, with a male predominance [1,4]. The course of visual loss is usually acute or subacute. The optic disc becomes hyperemic and is associated with peripapillary telangiectasia. The retinal nerve fiber layers are swollen. Over months, the disc edema subsides and becomes pale and atrophic. Visual acuity deteriorates to the degree of less than 20/200, commonly accompanied by a cecocentral scotoma [1]. Both eyes are involved with or without intervals [1]. Three primary mitochondrial DNA (mtDNA) mutations underlie the main pathogenesis of LHON. The first association of a mtDNA 11778 mutation with LHON was reported by Wallace and colleagues in 1988 [5]. These three primary mutations of mtDNA 11778, 14484, and 3460 encodes the NADH dehydrogenase subunit 4, subunit 6, and subunit 1 of Complex I of the respiratory chain, respectively [5,6,7,8]. 

Despite intense studies on the clinical and molecular aspects of LHON from 1988, the pathogenesis is still unclear, especially in the area of gender prevalence and penetrance. The mtDNA mutation is transmitted to all maternal lineages. However, only approximately 50% of men and 10% of women harboring a pathogenic mtDNA mutation develop optic neuropathy, reflecting both the incomplete penetrance and the gender prevalence difference [1]. LHON is well-known for its male prevalence, where over 80% of patients are male [1,2]. This male predominance exists in different mutation groups with a male to female ratio of 3:1, 4–6:1, and 8:1 in patients harboring the 3460, 11778, and 14484 mutations, respectively [1,4]. The reason for this male predominance remains unknown. 

The penetrance in LHON is incomplete and variable, where a positive family history was found in 50% of patients with the 11778 mutation, 71% with the 3460 mutations, and 100% with the 14484 mutation [1,9]. The penetrance of LHON is variable, even with the same mutation in homoplasmic fashion within the same family in a different pedigree branch [1,10]. All of these features cannot be explained by a single point mutation of mtDNA alone. Thus, genetic and epigenetic factors have been presumed to be involved in the penetrance of the LHON. Previously investigated genetic modifiers include heteroplasmy [11,12,13,14], secondary mtDNA mutations [15,16], mtDNA haplogroup [17,18,19], X-linked modifying gene or susceptibility locus [20,21,22], and other nuclear genes [23]. Tobacco and alcohol consumption have been proposed to be related to the onset of LHON, but the results are controversial from different studies [24,25]. 

Whole exome sequencing (WES) is a molecular technique for sequencing the protein-coding region of all genes in a whole genome. As humans have about 180,000 exons, constituting about 1% of the human genome, to sequence the exonic DNA, one needs to use high-throughput next generation DNA sequencing technology. WES may help to identify genetic mutations that alter protein sequences in either Mendelian and common polygenic diseases. It could be performed at a much lower cost than whole-genome sequencing. Thus, whole exome sequencing has been used in different fields of basic research and clinical practice [26].

The pathogenesis of incomplete penetrance and male prevalence in LHON is still unclear [1]. The comparative genomic hybridization (CGH) technique detects the chromosomal copy number changes. With the CGH technique, we did not detect any chromosomal abnormalities in LHON patients or in unaffected relatives [27]. The male predominance suggests that an X-linked modifying gene acts in concert with the pathogenic mtDNA mutation [20,22]. However, many other previous studies have failed to support an X-linked susceptibility locus based on linkage analysis [10], X-inactivation analysis or skewed X-inactivation in the affected tissues [28], or meiotic breakpoint mapping [29]. There have been many other studies investigating LHON susceptibility nuclear genes such as NDUFA-1, EPHX1…etc. [30,31] by using conventional genetic approaches. However, no conclusive evidence supports the pathogenic role of these genetic candidates. 

With the new technology of whole exome sequencing (WES), Jiang PP et al. searched the nuclear modifier gene in a LHON family with four family members, and presumed YARS2 as the candidate gene [32]. YARS2 gene is yet to be verified by further study for its role in the LHON penetrance, since it came from a single family only. 

In this study, we aimed to identify the LHON susceptibility allele(s) in the nuclear genomes by using whole exome sequencing technology. We intended to examine multiple LHON families including the LHON proband (11778 G > A mutation), adult sibling carrier, and the proband’s mother. By taking advantage of a larger cohort with multiple LHON families, we might be able to find the nuclear modifier gene responsible for the disease penetrance.

## 2. Results

A total of five families with the mt11778G > A mutation including seven patients and 10 relative carriers were enrolled in this study (Figure 1B).

The mean age of the LHON patients was significant younger than the carrier group (mean age, 30.71 years in the LHON vs. 50.40 years in the carrier group, *p* = 0.005). There was a significant male predominance in the LHON group compared to female predominance in the carrier group (male percentage, 86% in the LHON vs. 20% in the carrier group, *p* = 0.01). The LHON patients had significantly worse BCVA compared to the carrier group (mean logMAR, 1.80 in the LHON vs. 0.02 in the carrier, *p* < 0.001). Regarding the other ocular examinations, the LHON patients had worse VF and a thinner peripapillary RNFL and GCIPL thickness on OCT compared to the carrier group (all *p* < 0.05) (Table 1). 

### 2.1. Identification of Nuclear Modifier Genes 

To identify the nuclear modifier gene for the phenotypic penetrance of the mtDNA 11778 mutation associated LHON, we performed exome sequencing of DNA from five families with 17 members. These samples were divided into the proband group (probands with acute onset of LHON, *n* = 7) and control group (carriers including mother and relative carriers with the mtDNA 11778 mutation, without clinical manifestation of LHON, *n* = 10).

We found that many MT genes had a high percentage of variants in either the proband group or control group, with missense mutation as the major mutation type (Figure 2). Based on the gene burden method, we calculated the affected sample rate of genes for the control and proband groups. Then, we compared the difference of the affected sample rate between the control and proband groups. We focused on the top difference and a short list of genes due to these genes providing more correlations and more potential to be a key regulator between the control and proband groups. The difference of 0.3 gave us around ten genes, which is in accordance with our expectations. Therefore, the MT genes with a difference over 0.3 of mutation percentage between the proband and control groups are shown in Table 2. 

We next asked whether these genes were involved in any pathway. To answer this question, we conducted the pathway analysis for the genes in Table 2. Notably, the pathway analysis revealed that these genes were associated with the cofactor metabolism pathways (Table 3) [33]. Collectively, these data suggest a difference in the mutation burden of MT genes between the controls and probands, underlining the cofactor metabolism pathways. The complete list of variants in this study are available from the corresponding author, A.-G.W., upon reasonable request.

### 2.2. Family-Based Analysis for MT Genes 

Next, we conducted family-based analysis and performed intra-family genotype filtering analysis. After the initial analysis, an inter-family comparison was conducted. We found several interesting candidate MT genes, but conflicting results existed in different families (Table 4). METAP1D (c.41G > T), ACACB (c.1029del), ME3 (c.972G > C), NIPSNAP3B (c.280G > C, c.476C > G), and NSUN4 (c.4A > G) seem to be involved in the penetrance of LHON.

### 2.3. GWAS (Genome Wide Association Study) Analysis

A GWAS (genome wide association study) was performed with the strategy of the Cochran–Armitage trend test. The Cochran–Armitage trend test was used with the proband versus the control as having a “trend”, which depends on the count of the minor/alternate allele D/A, which is zero within genotype dd/rr, one within genotype Dd/Ar, and two within genotype DD/AA. We found that ADGRG5 (Chr16:575620A:G), POLE4 (Chr2:7495872T:G), ERMAP (Chr1:4283044A:G), PIGR (Chr1:2069357C:T;2069358G:A), CDC42BPB (Chr14:102949A:G), PROK1 (Chr1:1104562A:G), BCAN (Chr 1:1566582C:T), NES (Chr1:1566698A:G,1566705T:C, 1566707T:C) may be involved in the clinical penetrance of LHON (Table 5).

## 3. Discussion

LHON is the major hereditary optic neuropathy in Taiwan [1]. It has a minimum point prevalence for the mtDNA LHON mutation of 11.82 per 100,000 subjects and the minimum point prevalence of visual failure due to LHON of 3.22 per 100,000 subjects in adults under 65 years of age in Northeast England [1,34]. It may cause bilateral blindness in a young adult and cause severe disability. Thus, it is of utmost importance to understand this disease. Even though it has not been difficult to diagnose since the development of molecular diagnosis, there are few treatments available for this disease. 

The incomplete penetrance and male prevalence are still the major unexplained issues in LHON. Nuclear modifier genes have been presumed to affect the penetrance of LHON-associated mtDNA mutations. Through whole exome analysis, we found that many mitochondria related (MT-related) genes had a high percentage of variants in either the proband group or control group (Figure 2). Forty percent of the top 10 mutated genes in the control group were the same as in the proband group, which may be related to a similar genetic background, since our controls were the proband’s relatives. However, all of the participants in the control group did not have an abnormal phenotype at the current stage. Therefore, the impact of these variants on the participants in the control group was not determined and need further follow-up. The MT genes with a difference over 0.3 of mutation percentage between the proband and control groups include AK4, NSUN4, RDH13, COQ3, and FAHD1. AK4 (Adenylate Kinase 4) encodes the enzymes of the adenylate kinase family, which localize to the mitochondrial matrix [35]. It is related to the metabolism of nucleotides, which can catalyze the reversible transfer of the phosphate group among the adenine and guanine nucleotides (www.genecards.org, accessed date: 28 July 2022) [35,36,37,38,39]. NSUN4 (NOP2/Sun RNA Methyltransferase 4) encodes 5-methylcytosine rRNA methyltransferase NSUN4 in humans, which methylates mitochondrial 12S rRNA and is probably involved in mitochondrial ribosome small subunit (SSU) maturation and mitochondrial ribosome assembly [36,40,41]. RDH13 (Retinol Dehydrogenase 13) encodes a mitochondrial short-chain dehydrogenase/reductase, which localizes at the entrance to the mitochondrial matrix [42]. It may aid retinoic acid production and protect the mitochondria against oxidative stress [35,36,42]. COQ3 (Coenzyme Q3, methyltransferase) encodes mitochondrial ubiquinone biosynthesis O-methyltransferase, which catalyzes the 2 O-methylation steps in the ubiquinone biosynthetic pathway [36,40,43]. FAHD1 (Fumarylacetoacetate Hydrolase Domain Containing 1) encodes mitochondrial aC.-Y.L.pyruvase FAHD1, which is able to hydrolyze fumarylpyruvate and acetylpyruvate in vitro [36,40,44,45]. It may also have oxaloacetate decarboxylase activity in eukaryotes [46]. Pathway analysis also supports the findings that these genes are associated with cofactor metabolism pathways (Table 3). As above-mentioned, AK4 encodes adenylate kinases, which catalyze the phosphorylation of AMP while using ATP/GTP as phosphate donors [35,36,37,38,39]. It may also be involved in thiamine metabolism [47,48,49]. RDH13 encodes a mitochondrial short-chain dehydrogenase/reductase that may protect the mitochondria against oxidative stress [35,36,42]. COQ3 encodes mitochondrial ubiquinone biosynthesis O-methyltransferase and is involved in the ubiquinone biosynthetic pathway, while ubiquinone is an important electron carrier in the mitochondrial respiratory chain [36,40,43]. All of these genes contribute to the biosynthesis of cofactors and help to maintain the essential functions of mitochondria. The malfunction of these genes may endanger the essential functions of mitochondria including the respiratory chain reaction, which is important for the pathophysiology of LHON.

Family-based analysis showed that several candidate MT genes including METAP1D (c.41G > T), ACACB (c.1029del), ME3 (c.972G > C), NIPSNAP3B (c.280G > C, c.476C > G), and NSUN4 (c.4A > G) seem to be involved in the penetrance of LHON (Table 3). METAP1D (methionyl aminopeptidase type 1D, mitochondrial) is a gene encoding the mitochondrial aminopeptidase responsible for removing the N-terminal methionine from many proteins [50]. METAP1D gene mutation has been associated with spinocerebellar ataxia (www.genecards.org, accessed date: 28 July 2022) [36]. ACACB (acetyl-CoA carboxylase beta) is a biotin-containing enzyme that catalyzes the carboxylation of acetyl-CoA to malonyl-CoA, the rate-limiting step in fatty acid synthesis [36]. ME3 (malic enzyme 3) encodes an isoform of mitochondrial NADP (+)-dependent malic enzyme, which catalyzes the oxidative decarboxylation of malate to pyruvate using NADP+ as a cofactor [36]. NIPSNAP3B protein belongs to a vesicular trafficking-related protein family [51], and seems to be associated with type I 3-methylglutaconic aciduria and anemia of prematurity [36]. In family-based analysis, the results showed some differences between different families. However, there seemed to be a trend of dose-effect. For example, homozygous in the proband vs. heterozygous in the control and heterozygous in the proband vs. null in the control. Therefore, we still believe that these variants may contribute to the penetrance of LHON.

A GWAS (genome wide association study) was performed, which found that ADGRG5 (Chr16:575620A:G), POLE4 (Chr2:7495872T:G), ERMAP (Chr1:4283044A:G), PIGR (Chr1:2069357C:T;2069358G:A), CDC42BPB (Chr14:102949A:G), PROK1 (Chr1:1104562A:G), BCAN (Chr 1:1566582C:T), and NES (Chr1:1566698A:G,1566705T:C,1566707T:C) may be involved (Table 4). ADGRG5 (adhesion G protein-coupled receptor G5) encodes a member of the adhesion family of G-protein coupled receptors. They may play a role in the immune system as well as in the central nervous system [36]. POLE4 (DNA polymerase epsilon 4, accessory subunit) is a histone-fold protein that interacts with other histone-fold proteins to bind DNA in a sequence-independent manner [36]. ERMAP (erythroblast membrane associated protein) is a cell surface transmembrane protein that may act as an erythroid cell receptor, possibly as a mediator of cell adhesion [36]. PIGR (polymeric immunoglobulin receptor) is a member of the immunoglobulin superfamily [36]. CDC42BPB (CDC42 binding protein kinase beta) encodes a member of the serine/threonine protein kinase family containing a Cdc42/Rac-binding p21 binding domain resembling that of PAK kinase [36]. The PROK1 (prokineticin 1) protein may induce proliferation, migration, and fenestration (the formation of membrane discontinuities) in the capillary endothelial cells derived from endocrine glands [41]. BCAN (brevican) encodes a member of the lectican family of chondroitin sulfate proteoglycans, specifically expressed in the central nervous system. It is developmentally regulated and may function in the formation of the brain extracellular matrix [36]. NES (nestin) encodes a member of the intermediate filament protein family and is expressed primarily in nerve cells [36].

Although the clinical studies disclose a male predilection in LHON, we did not find a significant change in sex chromosome in our studies [1,52]. The mystery of gender bias in LHON may need more investigation to unravel.

This study was limited to the small sample size. However, we collected both the proband and carriers in the same family and established the family-based analysis. We believe that a family-based strategy could help to avoid the confounding factor from population stratification in genetic studies. In addition, we included only mitochondria-related nuclear genes for efficacious analysis, which may miss other genes that might also contribute to clinical penetrance. Moreover, we did not have a prediction model regarding the structural/functional alterations of the variants, and further investigations including verifying these candidate genes are required. Although it has many limitations, this study provides a new profile of candidate nuclear modifier genes for the clinical penetrance of LHON.

In summary, the incomplete penetrance and male prevalence are still the major unexplained issues in LHON. Through whole exome analysis, we found that several MT genes with a high percentage of variants were involved in a family-based analysis. A difference in the mutation burden of MT genes underlining the cofactor metabolism pathways was suggested by pathway analysis. In addition, the GWAS analysis also revealed several candidate nuclear modifier genes. The new technology of WES contributes to providing a highly efficient candidate gene screening function in molecular genetics. However, these candidate modifier genes need further verification for their modifying effect on the clinical penetrance in a larger cohort as well as more in-depth investigation of their biologic mechanism.

## 4. Materials and Methods

This prospective cross-sectional study was conducted from 6 January 2020 to 31 December 2021 at Taipei Veterans General Hospital. Eligible LHON patients with mitochondrial(mt) 11778 mutation G > A and their relatives who also carried the mt11778G > A mutation were invited to participate in this study consecutively. The study was approved by the institutional review board of Taipei Veterans General Hospital and the study protocol adhered to the tenets of the Declaration of Helsinki. Written informed consent was obtained from each participant after the goals and methods of the study were fully explained.

### 4.1. Families and Subjects

DNA samples used for this investigation were obtained from members of families who carried the mtDNA 11778G > A mutation including the probands, proband’s mother, and adult relative carriers. The genomic DNA of the probands and family members will be extracted from the peripheral blood for WES. Ophthalmic examinations and other clinical evaluations of probands and other members of these families were conducted. 

Subjects who did not have genetic-proved mtDNA 11778G > A mutation had no comparable relative carriers or who could not cooperate with the ophthalmic examinations were excluded from the study.

### 4.2. Ophthalmic Examinations

The best-corrected visual acuity (BCVA) was examined with the Snellen visual acuity chart. Other ophthalmic examinations included intraocular pressure, slit-lamp examination, fundoscopy, visual field (VF) with the Humphrey 30-2 SITA-standard protocol (Humphrey Field Analyzer 3, Carl Zeiss Meditec, Dublin, CA, USA), and optical coherence tomography (OCT) (Avanti RTVue XR, Optovue, Inc., Fremont, CA, USA) for the peripapillary retinal nerve fiber layer (RNFL) and ganglion cell-inner plexiform layer (GCIPL) thickness was conducted if feasible. 

### 4.3. Sample Collection and DNA Extraction

Peripheral blood (10 mL) was collected from each participant after obtaining their informed consent. DNA was extracted using Nucleospin Blood Mini (MACHEREY-NAGEL, Düren, Germany) according to the manufacturer’s instructions. Briefly, 20 µL Proteinase K and 200 µL blood were mixed, and 200 µL B3 buffer was added for 15 min of incubation at 70 °C. Next, 210 μL of ethanol was added to the lysate from the last step and mixed thoroughly. Then, the sample was transferred to the NucleoSpin^®^ Blood Column and centrifuged for 1 min at 11,000× *g* for DNA binding. For the silica membrane wash, 500 μL of Buffer BW was added to the column and centrifuged for 1 min at 11,000× *g* twice, followed by silica membrane dry spinning for 1 min at 11,000× *g*. The purified DNA was obtained by adding 100 µL preheated Buffer BE to the column and incubating for 1 min, followed by 1 min at 11,000× *g* centrifuge. 

### 4.4. Whole Exome Sequencing

Whole exome sequencing was performed according to the referenced paper as follows [53]. WES was conducted on 500 ng of genomic DNA from the probands and their family members. Fragment libraries were prepared from the sheared samples by sonication and target enrichment was performed according to the manufacturer’s protocols (Agilent SureSelect QXT ALL Human Exon V6 Kit or Roche KAPA HyperExome Kit). Captured DNA was amplified followed by solid-phase bridge amplification and paired-end sequenced on Illumina NovaSeq 6000 (Illumina, Inc., San Diego, CA, USA). Alignment of reads to the human reference sequence (hg38 assembly) and variant detection were performed using DRAGEN 3.7.5 (Illumina, Inc.) with the alt-aware configuration. The variant annotation information was obtained from Variant Effect Predictor (version 100) and Jannovar (version 0.35) with dbNSFP 4.1a. The novel variants were filtered against 1000 Genomes (1000 genomes release phase 3, http://www.1000genomes.org/, accessed date: 28 July 2022), dbSNP (http://www.ncbi.nlm.nih.gov/projects/SNP/snp_summary.cgi accessed date: 28 July 2022), and the Genome Aggregation Database (gnomad.broadinstitute.org accessed date: 28 July 2022) [54,55,56]. 

### 4.5. Mitochondria-Related Genes Analysis

The overview of the exome analysis is summarized as follows. First, we used the Msigdb mitochondria (MT)-related canonical pathway (CP) database with 838 genes and the MitoCarta3.0 database with 1135 genes to sort out a total of 1273 mitochondria-related nuclear genes (MT genes) as our target genes. Next, we excluded UTR, intron and synonymous variants, and also excluded those variants labeled as benign and likely benign in the ClinVar database. We filtered for variants including donor/splice acceptor, start loss, stop gained, frameshift, missense, and CADD Phred > 20/REVEL >0.5 (Figure 1A). Thereafter, we extracted the most significant variants from the analysis of all of the probands and controls. We also investigated the intra-family variants differences between the proband and carriers as our family-based analysis, which may help to find the variants important for penetrance for participants with the same genetic background (Figure 1A). The pathway analysis was performed using R package “clusterProfiler” with the ontology database of the Kyoto Encyclopedia of Genes and Genomes (KEGG) [47]. An adjusted *p* (Benjamini–Hochberg method) < 0.05 was considered as statistically significant for the pathway analysis.

### 4.6. GWAS Analysis

The input of GWAS was the exome sequencing data, which was converted to the PLINK format for the GWAS analysis. The GWAS analysis focuses on the correlation between the control and proband groups at a scale of SNP or the locus near SNP, while the gene burden analysis aims to look for candidate genes. We applied two different approaches intending to explore any genes or SNPs that potentially contribute to the pathogenesis of LHON. The gvcf files of all samples were combined and genotyped using the GATK bundle (4.2.0.0). The quality of cohort variants was evaluated by the Variant Quality Score Recalibration (VQSR) with the indel and SNP tranche of 99 and 99.8, respectively. The variants with GQ <20, depth of diploid <10, depth of haploid <5, allele fraction <0.2 and >0.8 for heterozygous or low inbreeding coefficient <−0.8 were excluded. The vcf format of the high-quality variants was converted to the PLINK (v1.90b6.24) bed format. Single SNPs had to meet the following criteria: minor allele frequencies (MAF) >10%, missing rate <10%, Hardy–Weinberg equilibrium (HWE) significance threshold <0.00001, otherwise, they were excluded from further analysis. The association analysis was conducted by using PLNK with the “--model” function. A *p* < 0.05 was considered as statistically significant. 

### 4.7. Family-Based Analysis

The MT genes obtained from Section 4.5 were also used for family-based analysis (Figure 1A). We applied an intra-family filter that looked for variants that were either homozygous alternate in the proband and heterozygous alternate in the control, or heterozygous alternate in the proband and homozygous reference in the control. Then, we counted the concurrence of matched variants across families and reported variants with a concurrence count >1.

## Figures and Tables

**Figure 1 ijms-23-11891-f001:**
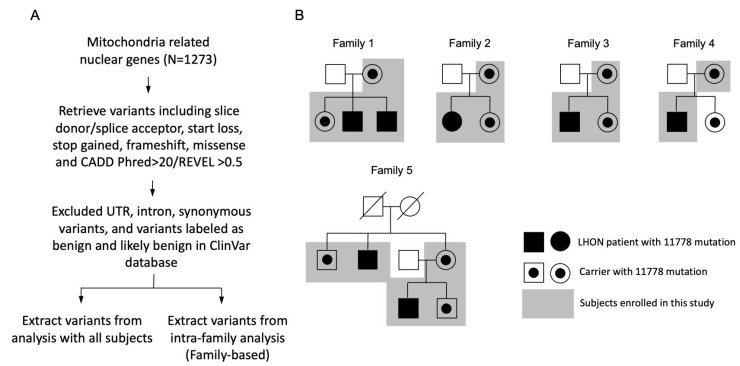
(**A**) Summary of exome sequencing in this study. (**B**) Five Chinese pedigrees with LHON. Vision-impaired individuals are indicated by blackened symbols.

**Figure 2 ijms-23-11891-f002:**
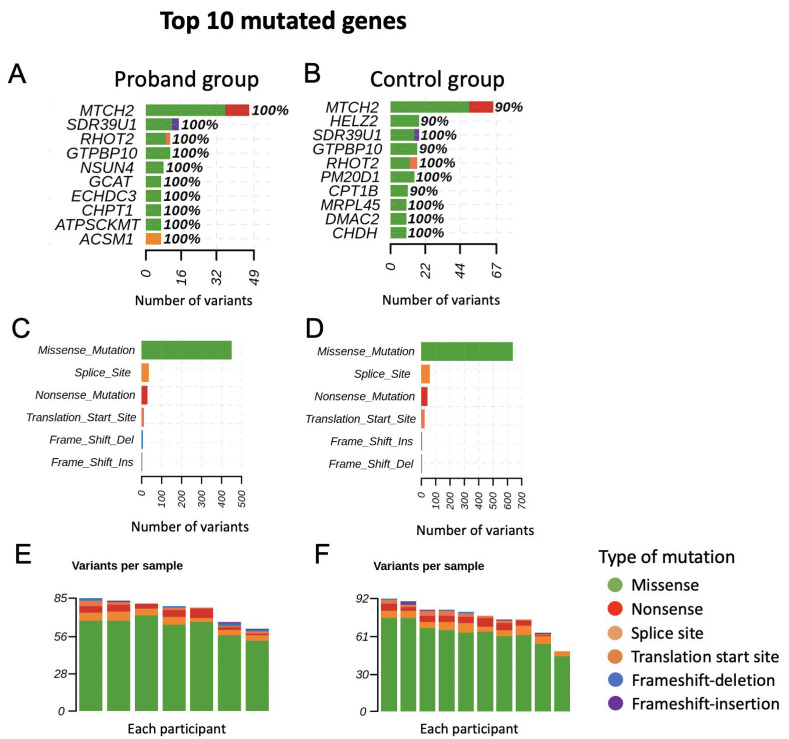
Mitochondria-related nuclear genes with variants in the proband and control group. The top 10 mutated genes are shown for the proband (**A**) and control group (**B**). The percentage indicates the ratio of mutation in all samples. Counts of consequences of variants for the proband (**C**) and control group (**D**). Counts of variants per sample for the proband (**E**) and control group (**F**). The color of the bars indicates the consequences of the variants while each bar presents a sample. We found that many MT genes had a high percentage of variants in either the proband group or control group, with missense mutation as the major mutation type. Forty percent of the top 10 mutated genes in the control group were the same as in the proband group, which may be related to a similar genetic background, since our controls were the proband’s relatives. The colors green, red, orange, international orange, sky blue, and violet indicate the missense, nonsense, splice site, translation start site, frameshift deletion, and frameshift insertion, respectively.

**Table 1 ijms-23-11891-t001:** The basic characteristics of patients with Leber’s hereditary optic neuropathy (LHON) and their relative carrier with the mitochondrial 11778 mutation.

	LHON Patient (*n* = 7)	Relative Carrier (*n* = 10) *	*p* Value ^#^
Age in years, mean (SD)	30.71 (11.32)	50.40 (12.87)	0.005
Gender			
Male, *n* (%)	6 (86%)	2 (20%)	0.01
Female, *n* (%)	1 (14%)	8 (80%)	
Best-corrected visual acuity			<0.001
logMAR, mean (SD) [range]	1.80 (0.68) [0.6~2.8]	0.02 (0.08) [−0.18~0.22]	
Snellen visual acuity [range]	[LP~0.3]	[0.6~1.5]	
RNFL thickness, mean (SD)	60.82 (11.69)	101.43 (5.92)	<0.001
GCIPL thickness, mean (SD)	50.42 (3.15)	76.33 (8.94)	<0.001
MD of VF in dB, mean (SD)	−17.19 (12.52)	−1.11 (2.08)	0.03

* One patient in the carrier group had poor BCVA due to cataract. ^#^
*t*-test for continuous variables and Fisher’s exact test for categorical variables. GCIPL, ganglion cell-inner plexiform layer; MD, mean deviation; logMAR, the logarithm of the minimum angle of resolution; RNFL, retinal nerve fiber layer; SD, standard deviation; VF, visual field.

**Table 2 ijms-23-11891-t002:** Mitochondria-related nuclear genes with aa difference over 0.3 of variant percentage between the proband and control groups.

Symbol	Chromosome	Variant Percentage		Differences
		Control	Proband	
AK4	chr1	0.70	0.29	−0.41
NSUN4	chr1	0.60	1	0.40
RDH13	chr19	0.50	0.14	−0.36
COQ3	chr6	0.50	0.86	0.36
FAHD1	chr16	0.10	0.43	0.33
CHPT1	chr12	0.70	1.00	0.30
METAP1D	chr2	0.70	1.00	0.30
MRM1	chr17	0.30	0	−0.30
NCOA6	chr20	0.30	0	−0.30
TOP3A	chr17	0.30	0	−0.30

Chr, chromosome.

**Table 3 ijms-23-11891-t003:** Pathway analysis of the mitochondria-related nuclear genes with a difference over 0.3 of variant percentage between the proband and control groups.

Pathway	Symbol	Adjusted *p* Value
Biosynthesis of cofactors *	AK4/RDH13/COQ3	0.000628845
Ubiquinone and other terpenoid-quinone biosynthesis	COQ3	0.030532282
Thiamine metabolism	AK4	0.030532282

* Biosynthesis of cofactors: A metabolism of essential cofactors used by the minimal cell starting with precursors such as nicotinamide, riboflavin, thiamine, pyridoxal, pantothenic acid, methionine, and folic acid [33].

**Table 4 ijms-23-11891-t004:** Family-based analysis of mitochondria-related nuclear genes between the proband and control groups.

Gene Symbol	Chromosome	Family	Description
METAP1D	Chr2	1	Heterozygous c.41G > T in proband
		2	Homozygous c.41G > T proband, heterozygous in control
ACACB	Chr12	1	Heterozygous c.1029del in 2 probands
		3	Missense variant found in control
		4	Heterozygous c.1951G > A in proband
ME3	Chr11	2	Heterozygous c.972G > C in proband
		4	Homozygous c.972G > C in proband, heterozygous in control
NIPSNAP3B	Chr9	3	Homozygous c.280G > C, c.476C > G in proband, heterozygous in control
		4	Heterozygous c.280G > C, c.476C > G in proband
NSUN4	Chr1	4	Homozygous c.4A > G in proband, heterozygous in control
		5	Heterozygous c.4A > G in proband

**Table 5 ijms-23-11891-t005:** The genome wide association study (GWAS) analysis of the proband and control groups.

Symbol	Gene ID	SNP	HGVSc	*p* Value
ADGRG5	221188	chr16:57562087:A:G	NM_001304376.3: c.-7A > G	0.001
POLE4	56655	chr2:74958729:G:T	NM_019896.4: c.50G > T	0.002
ERMAP	114625	chr1:42830447:G:A	NM_001017922.2: c.-2G > A	0.005
PIGR	5284	chr1:206935771:C:T	NM_002644.4: c.1093G > A	0.006
		chr1:206935822:G:A	NM_002644.4: c.1046-4C > T	0.006
CDC42BPB	9578	chr14:102949759:G:A	NM_006035.4: c.3449 + 6C > T	0.006
PROK1	84432	chr1:110456232:G:A	NM_032414.3: c.199G > A	0.007
BCAN	63827	chr1:156658204:T:C	NM_021948.5: c.2370T > C	0.008
NES	10763	chr1:156669844:G:A	NM_006617.2: c.4344C > T	0.008
		chr1:156670516:C:T	NM_006617.2: c.3672G > A	0.008
		chr1:156670711:C:T	NM_006617.2: c.3477G > A	0.008

HGVSc, Human Genome Variation Society notation in the cDNA; SNP, single nucleotide polymorphism.

## Data Availability

The data presented in this study are available on request from the corresponding author, A.-G.W. The data are not publicly available due to ethical restrictions.

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
