# Peer review of "Candidate Modifier Genes for the Penetrance of Leber’s Hereditary Optic Neuropathy"

_ijms, 2022, doi:10.3390/ijms231911891_

Round 1
Reviewer 1 Report
Excellent paper who demonstrates candidate modifier genes for the penetrance of Leber’s herediitary optic neuropathy. The authors conclude that technology of WES contributes to provide a highly efficient candidate gene screening function in the molecular genetics but also that these candidate modifier genes need further verification for their modifying effect on the clinical penetrance in 3 a larger cohort, and more in-depth investigation of their biologic mechanism.
Very novelty and very interesting. The methods used by the authors are very good. Good conceptualization. Excellent!
Author Response
Thank you for your kindly comment.

Reviewer 2 Report
As a clinical ophthalmologist I appreciate the work but I cannot judge the accuracy of some laboratory (genetic) tests.
The work will certainly contribute to further research into the nature of LHON.
Author Response
Thank you for your kindly comment.

Reviewer 3 Report
In this manuscript the authors profiled the whole exome of nuclear genes encoding for mitochondrial proteins in probands and carrier subjects in order to find candidate modifiers able to modulate the penetrance of Leber’s Hereditary Optic Neuropathy.
The study is of potential interest but the following points need to be addressed.
Major points
- The description of sample collection and DNA extraction is missing, please amend.
-In the main text only the top 10 mutated genes have been reported. Is there a repository where the authors have charged the complete list of variants or at least the ones with a predicted pathological significance? A "data availability" section containing this information should be included in the manuscript
-Page 5 line 31: what does it means “a difference over 0.3 of mutation percentage between proband and control groups"? Which is the rationale of that choice? Please clarify.
-Which are the input lists for pathway analysis and genome wide association study? Please briefly, describe also the rationale of that choice.
-The authors should describe which is the physiological significance of the variants found in the control groupì
-The caption of figure two is not exhaustive the panel should be described in more details. For example, what does the violet colour that appears in the upper panels refers to?
-In the intra-family analysis, the authors identified differences in variants between controls and probands. However conflicting results emerged when comparing different families, which makes it questionable to conclude that these variants may be involved in regulating the penetrance of LHON. How do the authors explain these discrepancies?
-Have the authors made any inference on the predicted structural/functional alterations derived from the main variants putatively involved in regulating the clinical penetrance of LHON? The authors should include a table to resume such information.
-Discussion section: It is unclear why there are repeated description of the function of the same genes see for example AK4, RDH13 and COQ3. Moreover, rather than describing in general the function of the candidate genes that emerged from the study, the authors should focus on how the altered function of the corresponding proteins could influence the course of the disease.
Minor points
-On page 2 lines 33 34 the authors state: "Tobacco and alcohol consumption were considered as epigenetic factors which may affect the penetrance as well 24-25", however "reference 24 and 25 do not refer to epigenetic modifications induced by tobacco and alcool consumption. In particular according to ref 25 no significant deleterious association between tobacco or and vision loss among individuals harboring LHON is detectable. Please clarify.
-Page 2 line 8: “..became pallor and atrophic”, maybe the authors mean “…became pale and atrophic”?
-Page 2 line : ”Vision deteriorated..”, maybe the authors mean ”Vision acuity deteriorated..”
-Page 2 lines 46-47: the authors state “The male predominance suggests an X-linked modifying gene act with the pathogenic mtDNA mutation”, maybe they mean “The male predominance suggests an X-linked modifying gene acts in concert with the pathogenic mtDNA mutation”?
-Page 4 lines 10-11: “We also investigated the intra-family variants differences between proband and 10 carriers as our family-based analysis, which may imply the key of penetrance”, the sentence is unclear, what does it means which may imply the key of penetrance? Please rewrite.
-Page 5 line 1: “The variants…” which variants are the authors referring to? The ones affecting the top mutated genes? All the variants identified? The pathological ones? Please specify
-Page line 5: “…and reported variants that concurrence count >1.” Maybe the authors mean :”…and reported variants with concurrence count >1” ?
-Page 8 line43: “It may also involve in thiamine metabolism”, maybe the authors mean “It may also be involved in thiamine metabolism”?
Author Response
Thank you for your kindly comments. The point-by-point response was in the attachment, please see the attachment.

Round 2
Reviewer 2 Report
I have no comments on the work already modified and submitted
Reviewer 3 Report
The authors have satisfactorily responded to all my comments.